# Zoom-In Neural Network Deep-Learning Model for Alzheimer’s Disease Assessments

**DOI:** 10.3390/s22228887

**Published:** 2022-11-17

**Authors:** Bohyun Wang, Joon S. Lim

**Affiliations:** Department of Computer Science, Gachon University, Sujeong-gu, Seongnam-si 13557, Gyeonggi-do, Republic of Korea

**Keywords:** AAL functional regions, Alzheimer’s disease, deep neural networks, metacognitive learning, discriminative regions of interest of Alzheimer’s disease, resting-state fMRI

## Abstract

Deep neural networks have been successfully applied to generate predictive patterns from medical and diagnostic data. This paper presents an approach for assessing persons with Alzheimer’s disease (AD) mild cognitive impairment (MCI), compared with normal control (NC) persons, using the zoom-in neural network (ZNN) deep-learning algorithm. ZNN stacks a set of zoom-in learning units (ZLUs) in a feedforward hierarchy without backpropagation. The resting-state fMRI (rs-fMRI) dataset for AD assessments was obtained from the Alzheimer’s Disease Neuroimaging Initiative (ADNI). The Automated Anatomical Labeling (AAL-90) atlas, which provides 90 neuroanatomical functional regions, was used to assess and detect the implicated regions in the course of AD. The features of the ZNN are extracted from the 140-time series rs-fMRI voxel values in a region of the brain. ZNN yields the three classification accuracies of AD versus MCI and NC, NC versus AD and MCI, and MCI versus AD and NC of 97.7%, 84.8%, and 72.7%, respectively, with the seven discriminative regions of interest (ROIs) in the AAL-90.

## 1. Introduction

Neuroscience has provided inspiration for and insight into artificial neural networks, including deep-learning networks. However, the field of artificial neural networks has been optimized through mathematics rather than neuroscientific findings [1]. In addition, there is a view that the backpropagation (backprop for short) rule itself is neurobiologically unrealistic [2]. Backprop networks are neuron-like supervised learning systems that adjust the synapses of the connections between neuron units [3]. However, whether and how this technology is implemented in the brain is still difficult to grasp [4].

Deep-learning algorithms with a hierarchical architecture are exploited as representative learning schemes, including restricted Boltzmann machines (RBMs), using local unsupervised pretraining layers from the bottom to top layers [5,6,7]. A deep belief network (DBN) is composed of a stack of RBMs, and full backpropagation fine-tuning is then performed. Recurrent neural networks (RNNs) are suitable for modeling sequential data [8]. A convolutional neural network (CNN) [9] consists of one or more subsampling convolutional layers, followed by a multilayer neural network through a supervised backprop. Deep Q-learning [10] demonstrates deep reinforcement learning trained by backprop in a deep-learning model. Although a backprop algorithm is mainly used in deep models, it causes obstacles in learning, such as a local minimum, slow convergence [11], and vanishing gradient problem [12].

A simple feedback alignment mechanism that adjusts weights by multiplying errors with even random synaptic weights performs as effectively as backprop [13]. The algorithm uses random weights instead of a symmetric backward connectivity pattern in backpropagation.

As one of the features of deep-learning algorithms used in unsupervised learning, the pretraining is determined by starting with random weights. Biological learning is more advantageous for the optimization of weight initialization due to large numbers of trials and errors. However, because backprop consumes considerable time to correct the weights after initialization, optimization using various weight initializations is difficult to achieve.

An intuitive way to consider diverse brain learning paradigms is to find another sup-pattern that is different from the main pattern. For example, ambiguous forms for a particular pattern can be learned through a more detailed look using a magnifying glass. This allows us to make more accurate judgments by learning the causes from a misjudgment, prejudice, or misclassification. In other words, learning can come from mistakes that are to be discarded or ignored. At the same time, it is possible to learn more precisely only through the refined instances. This paradigm reinforces learning efficiency through rewards, from both mistakes and well-learned domains simultaneously.

According to Wikipedia, the dictionary meaning of metacognition is an awareness of one’s thought processes and an understanding of the patterns behind them. When metacognition is applied to a person, it means that the person understands his or her learning process or judgment process or thinking process. This concept was applied to the learning algorithm in this paper. We propose a feedforward zoom-in neural network (ZNN) deep-learning algorithm that implements a learning paradigm called metacognitive learning. A ZNN stacks layers, each of which consists of a set of zoom-in learning units (ZLUs) in a feedforward manner. In a ZLU, meta-cognitive information is accumulated through subpattern learning and fine-tuning learning according to the learning results after learning occurs. Through subpattern learning, another pattern is found as meta-cognitive information from errors in standard learning. Through fine-tuning learning, a more detailed pattern for standard learning is found as meta-cognitive information.

The outputs of the ZLUs in a layer are forwarded to a higher layer to achieve a gradual improvement in the discrimination power. This brings about a dimensionality reduction with structural flexibility by decreasing the number of ZLUs as the learning of the ZNN moves to higher-level layers [14].

## 2. Zoom-In Neural Network with Subpattern and Refined Learning

In this paper, a deep-learning model based on a neural network with weighted fuzzy membership functions (NEWFMs) [15] is proposed. The NEWFM is a supervised feedforward neuro-fuzzy system that uses the bounded sum of weighted fuzzy membership functions (BSWFMs) for classification. The structure of the NEWFM is composed of three layers: the input layers, the hyperbox layers, and the output layers. The NEWFM reinforces BSWFMs in the hyperbox layer for training. The learning scheme adjusts each BSWFM assigned to each feature to construct a BSWFM. The output values from the output layer can be defuzzified by the Takagi–Sugeno method [16]. The Takagi-Sugeno method is fuzzy is a type of fuzzy model that generates defuzzified values in a non-linear manner.

### 2.1. Structure of Zoom-In Neural Network

The proposed ZNN is a supervised feedforward deep-learning system using ZLUs implemented by the NEWFMs. The structure of the ZNN, illustrated in Figure 1, is composed of an input layer, multi-ZLU layers, and an output layer. The input layer is split into subsets of input features according to their characteristics [17]. Each ZLU in a ZLU layer performs standard learning followed by a supervised subpattern and refined learning to generate an output to the next ZLU layer or output layer. The main contribution and advantage of a ZNN is that it learns without a backpropagation or vanishing gradient problem and uses metacognitive information during learning process. This is because every ZLU layer executes a supervised subpattern and refined learning for the next ZLU layer.

### 2.2. Zoom-In Learning Unit Processes

The basic processes of the ZNN are executed by the neuronal ZLU module. Each ZLU generates two subpatterns and a refined defuzzified output for the next layer. The subpatterns and the refined defuzzified output include metacognitive information.

A ZLU consists of two-part learning, as shown in Figure 2. The bottom NEWFM is trained for the standard training process from the split input features. Then, the input instances are classified into two groups of misclassified instances (MIs) and correctly classified instances (CCIs) by the bottom NEWFM, which is called instance grouping in Figure 2. The top two NEWFMs execute subpattern and refined training by the MIs and CCIs, respectively, called zoom-in training, to find a subpattern of the MIs and a refined pattern of CCIs precisely. These patterns are used as metacognitive information. The subpattern training using MIs attempts to find another pattern ignored from the standard training. In contrast, the refined training learns in detail through the noise-reduced CCIs. For the test process after training, the input instances of ZLU are input to the top two NEWFMs directly to produce the output for the next layer.

The first advantage of a ZLU module is that the load of learning can be distributed to the ZLUs according to the divided input features. Second, a ZLU module can be assembled flexibly to find a pattern of functional connectivity of neural networks. As the third advantage, the cost function for each ZLU module could first be applied locally, which may follow the cognitive processes of the human brain learning inferences of a hierarchical principle by fusing or filtering out of the local decisions [18].

As the main contribution of ZLU, it applies a zoom-in learning process. A neuroscience basis for the learning scheme in the human brain is provided by the metacognitive learning method that monitors uncertainty regarding a decision [19,20], as in the standard training of a ZLU followed by revising the decision in ZLU zoom-in training.

### 2.3. Zoom-In Neural Network Algorithm

ZLU is a basic unit module of the ZNN algorithm. Each ZLU module generates two types of subpatterns and refined outputs for the next ZLU layer. The layerwise supervised learnings in ZNN are executed to improve the learning ability by zooming in on the input patterns using the ZLUs. Feedforward learning of a ZNN without backpropagation in deep-learning algorithms provides an efficient learning method by eliminating the vanishing gradient problem.

The ZNN algorithm is divided into training and testing parts, i.e., a training algorithm and test algorithm of the ZNN, respectively. Algorithms 1 and algorithm 2 show train and test algorithms. To describe the training and testing that takes place in a ZLU, once the standard training, instance grouping, and zoom-in training are completed in a ZLU, as indicated by the solid lines in Figure 2, the test will only run the zoom-in test along the dotted lines in Figure 2. Based on the feature selection function built in a NEWFM, the features can be selected during the standard training in a ZLU to reduce the dimensionality of the input of every ZLU layer. The output of the j-th ZLU in the i-th layer is represented by Takagi–Sugeno defuzzifications (TSDs) {Ti,j,M} and {Ti,j,C} for a subpattern and refined output, respectively. The detailed processes of the ZNN algorithm are as follows (Algorithms 1 and 2):


**Algorithm 1: *Train Algorithm of ZNN***
1 *Input Layer* 1.1 **Split** train input data with *n* features **into** the *k* subsets with   *n*/*k* features such that {*I*_0,1_}, {*I*_0,2_}, …, {*I*_0,*k*_}2 *i-th ZLU layer (initially i = 1)* 2.1 Let *ZLU*_*i,j*_ be *j*-th ZLU in *i*-th ZLU layer 2.2 **For each** {*I*_*i*−1,*j*_}, where *j* = 1 to *k*  2.2.1 **Assign** {*I*_*i*−1,*j*_} to *ZLU*_*i,j*_ as input 2.3 **For each**
*ZLU*_*i,j*_, where *j* = 1 to *k*  2.3.1 **Standard training**   2.3.1.1 **Standard training** using NEWFM_*i,j*_ from {*I*_*i*−1,*j*_}   2.3.1.2 **Instance grouping**: **divide** {*I*_*i*−1,*j*_} instances into      misclassified instances (MI) {*M*_*i,j*_} and correctly      classified instances (CCI) {*C*_*i,j*_}  2.3.2 **Zoom-in training**   2.3.2.1 **Subpattern training** from {*M*_*i,j*_} using      NEWFM_*i,j,M*_   2.3.2.2 **Refine training** from {*C*_*i,j*_} using NEWFM_*i,j,C*_  2.3.3 **Zoom-in output** 2.3.3.1 **Output** TSD {*T*_*i,j,M*_} using NEWFM_*i,j,M*_ from {*I*_*i*−1,*j*_} 2.3.3.2 **Output** TSD {*T*_*i,j,C*_} using NEWFM_*i,j,C*_ from {*I*_*i*−1,*j*_} 2.4 **Split** the TSDs {*T*_*i*,1,*M*_, *T*_*i*,1,*C*_, *T*_*i*,2,*M*_, *T*_*i*,2,*C*_, …, *T*_*i,n,M*_, *T*_*i,n,C*_}   **into** {*I*_*i*,1_}, {*I*_*i*,2_}, …, {*I_i,n_*}, where *n* is the number of ZLUs   in the (*i* + 1)-th ZLU layer 2.5 *i* = *i* + 1, *k* = *n*, and **go to** 2 **until** predefined *i* is reached3 *Output Layer* 3.1 **Train** by the NEWFM_*i*_ using {*I*_*i*−1,1_}, {*I*_*i*−1,2_}, …, {*I*_*i*−1,*k*_}


**Algorithm 2: *Test Algorithm of ZNN***
1 *Input Layer* 1.1 **Split** test input data with *n* features **into** the *k* subsets with   *n*/*k* features such that: {*I*_0,1_}, {*I*_0,2_}, …, {*I*_0,*k*_} as in 1.1 of*   **Train Algorithm of ZNN***2 *i-th ZLU Layer (initially i = 1)* 2.1 **For each** {*I*_*i*−1,*j*_}, where *j* = 1 to *k*  2.1.1 **Assign** {*I*_*i*−1,*j*_} to *ZLU*_*i,j*_ as input  2.1.2 **Zoom-in test:**   2.2.2.1 **Output** TSD {*T*_*i,j,M*_} using NEWFM_*i,j,M*_ from {*I*_*i*−1,*j*_}   2.2.2.2 **Output** TSD {*T*_*i,j,C*_} using NEWFM_*i,j,C*_ from {*I*_*i*−1,*j*_} 2.2 **Split** the TSDs {*T*_*i*,1,*M*_, *T*_*i*,1,*C*_, *T*_*i*,2,*M*_, *T*_*i*,2,*C*_, …, *T*_*i,n,M*_, *T*_*i,n,C*_}   **into** {*I*_*i*,1_}, {*I*_*i*,2_}, …, {*I*_*i,n*_}, where *n* is the number of ZLUs   in the (*i* + 1)-th ZLU layer 2.3 *i* = *i* + 1, *k* = *n*, and **go to** 2 **until** the *output layer* is reached3 *Output Layer* 3.1 **Output** TSD {*T*_*i*_} by the NEWFM_*i*_ using {*I*_*i*−1,1_}, {*I*_*i*−1,2_}, …,   {*I*_*i*−1,*k*_}

## 3. Experimental Results

In this section, the experimental results for assessing persons with Alzheimer’s disease (AD) and mild cognitive impairment (MCI), compared with normal control (NC) persons, using the proposed ZNN deep-learning algorithm, are presented to evaluate the ZNN algorithm with the discriminative regions of interest in the Anatomical Automatic Labeling (AAL-90) model.

### 3.1. Dataset

The experimental dataset contains the resting-state functional magnetic resonance imaging (rs-fMRI) data concerning 34 AD patients, 89 MCI patients, and 45 NC persons. It was obtained from the Alzheimer’s Disease Neuroimaging Initiative (ADNI) database [21]. The rs-fMRI dataset for each subject consisted of three-dimensional brain images with a 140-time series of voxels. Based on the AAL-90 model from 27 high-resolution T1-weighted images of a young male [22], the brain image was divided into 90 functional regions of interest (ROIs).

To obtain an overall change in each ROI with 2000 to 3000 voxels, the average of voxels (AOV) of each ROI was generated as a representative signal of the ROI. The features were extracted using the Haar wavelet transform (HWT) from the 140-time series of AOVs. In addition, the local and global graph measures [23] were used, as shown in Table 1.

### 3.2. Experimental Structure for Alzheimer’s Disease Assessments Using ZNN

The series of experimental structures led to a 140-time series of ROIs, AOVs, feature extraction, feature selection, region selection, and ZNN classification, as shown in Figure 3.

From the 140-time series of AOVi of ROIi (Figure 3a,b), the 32 HWT features of the d3, d4, and a4 coefficients and the 10 graphic features (Table 1) were extracted (Figure 3c). Then, 20 features were selected from among the 42 features extracted by the NEWFMi model (Figure 3d), producing classification accuracies for each ROIi.

Finally, the highly accurate ROIs that were candidates for discriminative ROIs for AD assessments were selected for input into the ZNN model (Figure 3e,f). In practice, the 16 best ROIs, each having 20 features, were the final inputs of the ZNN.

AD assessments by the proposed ZNN model have three pairs of classifications: AD versus NC-MCI (A-NM), NC versus AD-MCI (N-AM), and MCI versus AD-NC (M-AN). The ZNN model was run with a hold-out verification method by setting 23 subjects in the dataset as training sets and the rest (11 AD, 66 MCI, and 22 NC subjects) as test sets.

Figure 4 shows the ZNN model for Alzheimer’s disease classifications implemented from inputs of the 16 selected ROIs. The first ZLU layer was composed of 16 ZLUs that each received 20 features from an ROI; then, they were output to the second ZLU layer. Each ZLU in the first ZLU layer produced TSD {T1,j,M} and TSD {T1,j,C} as the second ZLU layer input. Then, the processes were recurrently executed in the second ZLU layer. Finally, the NEWFM of the output layer carried out an AD assessment using the second ZLU layer outputs. All of the detailed classification processes were performed according to the training algorithm of the ZNN and the test algorithm of the ZNN, as described in the previous section.

### 3.3. AD, MCI, and NC Classifications

ZNN yielded the three classification accuracies of A-NM, N-AM, and M-AN as 97.7%, 84.8%, and 72.7%, respectively, as shown in Table 2. Adding the layers of a ZNN, the accuracies were improved in cases of the three classifications, as shown in Table 3.

The increases in accuracy with the stacking of ZLU layers are shown in Figure 5. On an average, the second ZLU layer increased by 7.5% over the first ZLU layer, and the output layer improved by 16.3% over the second ZLU layer.

Ref. [33] shows 96.85% as the results of classifying AD and SCI using CNN. Although there are differences in the experimental environment, the comparison is meaningful in that the same ANDI data were used.

The 16 most prominent ROIs in the AAL-90 were selected in accordance with the accuracies of the ROIs in Figure 3d. The 16 ROIs were selected to operate a ZNN for each classification as an input layer in Figure 4. The final 16 ROIs for the three AD classifications of a ZNN are listed in Table 4.

Some of the ROIs in a column in Table 4 are also repeated in the other columns. The seven repeated discriminative ROIs in bold are defined as cingulum_mid_r, caudate_l, parietal_sup_l, frontal_mid_r, frontal_mid_l, parietal_inf_l, and postcentral_r for the AD assessments.

The most discriminating ROI in Table 4 is cingulum_mid_r, which plays a remarkable role in the progressive development of MCI and AD [34,35]. The eigenvector centrality shows significant differences in caudate_l between NC and AD [36]. In addition, the subcortical region of the caudate plays a key role in MCI and AD [37]. Other analyses reveal that effective connectivity from the right middle frontal gyrus to the left superior parietal (parietal_sup_l) cortex, as well as from the right to the parietal_sup_l gyrus, is decreased in prodromal AD patients [38]. AD patients atrophy in AD-specific regions related to cognitive performance in the caudal middle frontal gyrus related to frontal_mid_r and frontal_mid_l [39]. Functional connectivity differences of the postcentral gyrus are affected in early-onset AD within the sensory-motor system in the default-mode network [40]. As found in the references, the seven discriminative ROIs obtained by the chosen ZNN were validated as biomarkers for AD assessments.

Figure 6 shows the location of seven overlap areas of discriminative ROIs in the AAL-90 from the three AD classifications. From the experiments, we can see that the seven areas have important relations in the experiment and classification results. The BrainNet Viewer software package (URL: http://nitrc.org/projects/bnv accessed on January 2020) was used to create the images of the ROIs.

## 4. Discussion

In this paper, we proposed a deep neural network model, ZNN, to classify Alzheimer’s diseases, which is a new feedforward deep-learning model using brain-inspired metacognitive learning. The ZNN, model without a gradient descending problem in backprop, simulates activity-dependent learning using ZLU units, such as the synaptic plasticity in the brain. A neuroscientific basis for the learning scheme of the metacognitive learning in the human brain [20] is implemented by the subpattern and refined learning in a ZLU. The absence of a backprop, together with metacognitive learning, are the main technical elements that make this paper original.

In addition, the ZLU units enhance the connectivity of the neural networks by connecting the block-type neural network units in a flexible manner. In the proposed model, by using 10 graphic features and h\Haar wavelet coefficients together as classification features, we obtained the effect that the regional characteristics of the brain were more clearly reflected in the classification through the Haar wavelet. The proposed model can also be used to evaluate other diseases. The features used in this method are features that can be extracted from images and numerical data. As the data used for the classification of diseases are mostly image or numerical data, it is possible to apply this model.

The proposed algorithm yields a 97.7% classification accuracy for AD versus SCI-MCI, 84.8% classification accuracy for SCI versus AD-MCI, and 72.7% classification accuracy for MCI versus AD-SCI, using the discriminative ROI specifications in the AAL-90. Thus, ZNN learns more efficiently through the zoom-in learning processes following a human-like approach in the way it acts. A further matter to be considered is that AD and SCI are classifications in which symptoms are clearly distinguished. However, as MCI is an intermediate state between AD and SCI, the accuracy of the classification of MCI and others is lower than that of AD and SCI. Another document that shows that the MCI classification results are relatively low is [33].

The importance of classification of MCI is gradually increasing. Accurate classification of MCI enables early diagnosis or prevention of dementia. This can slow the progression of dementia, alleviate symptoms, or increase the probability of a cure. Therefore, improving the classification accuracy of MCI is a future research task.

One limitation in this study was that the number of data used was not balanced. We need to balance the data with additional up-to-date data from ANDI. Recent data are continuously accumulating on the ANDI site, and it is necessary to experiment with the accumulated data. We have one more limitation: As the datasets collected from ANDI were not all from the same machine [41], they may have contained errors in the measurements.

## Figures and Tables

**Figure 1 sensors-22-08887-f001:**
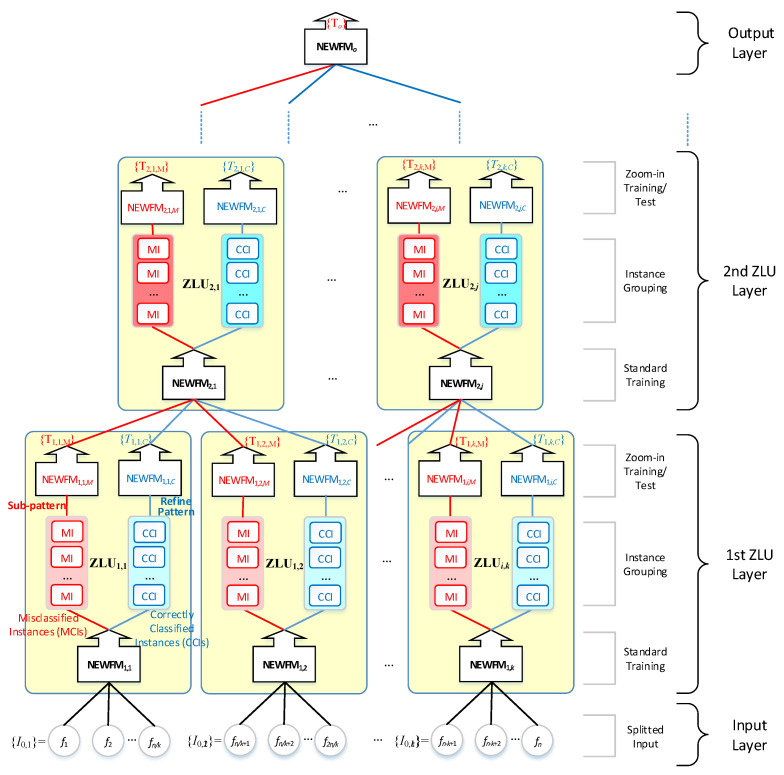
Structure of the Zoom-in Neural Network (ZNN).

**Figure 2 sensors-22-08887-f002:**
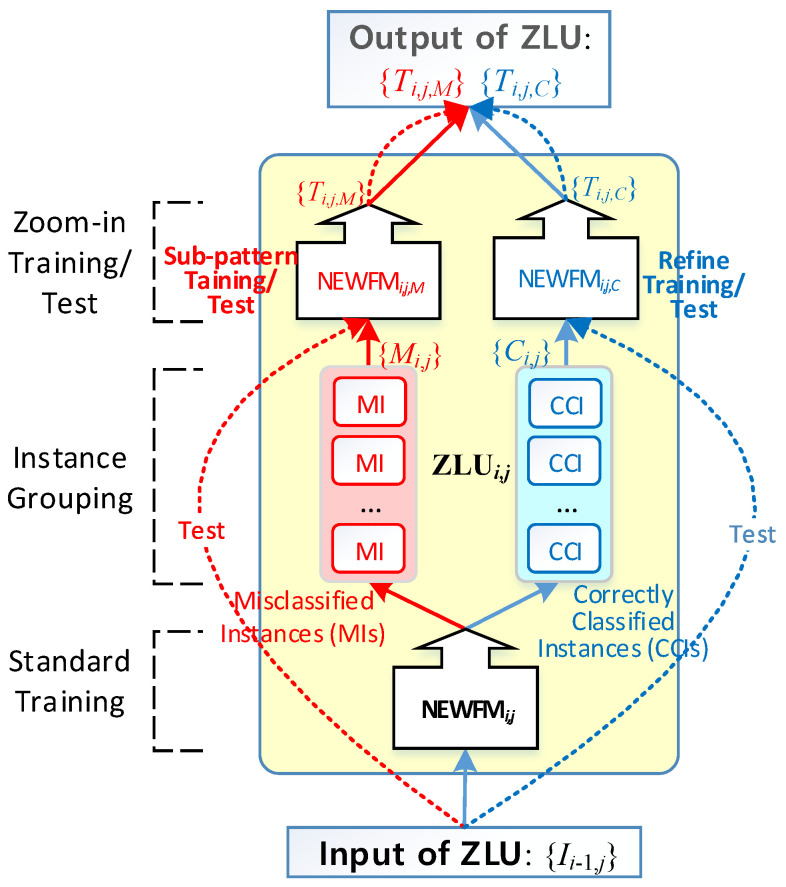
Structure of the Zoom-in Learning Unit (ZLU) (solid lines denote training and dotted lines denote tests).

**Figure 3 sensors-22-08887-f003:**
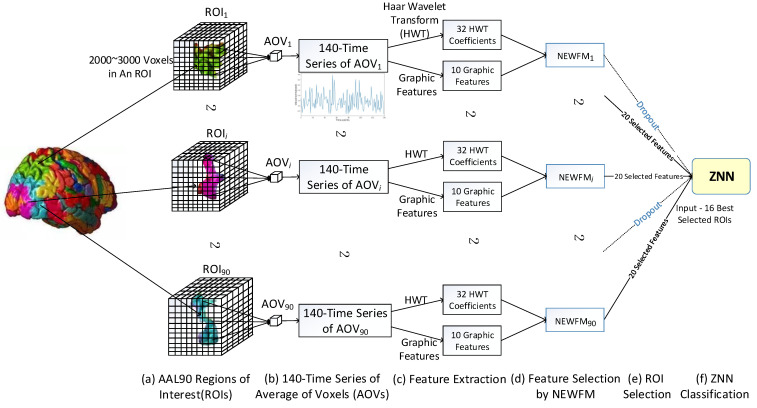
Schematic diagram for Alzheimer’s disease assessments.

**Figure 4 sensors-22-08887-f004:**
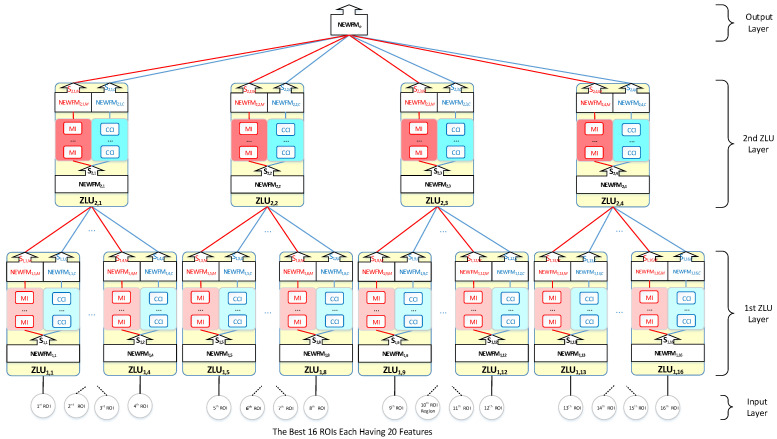
ZNN model for Alzheimer’s disease classification.

**Figure 5 sensors-22-08887-f005:**
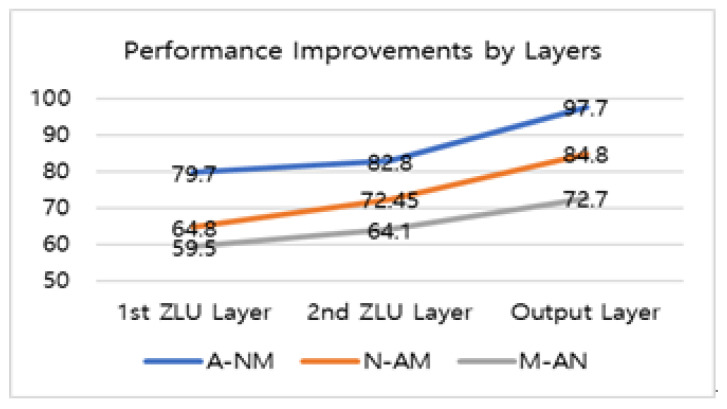
Performance improvements by stacking layers of the ZNN for three AD assessments (%).

**Figure 6 sensors-22-08887-f006:**
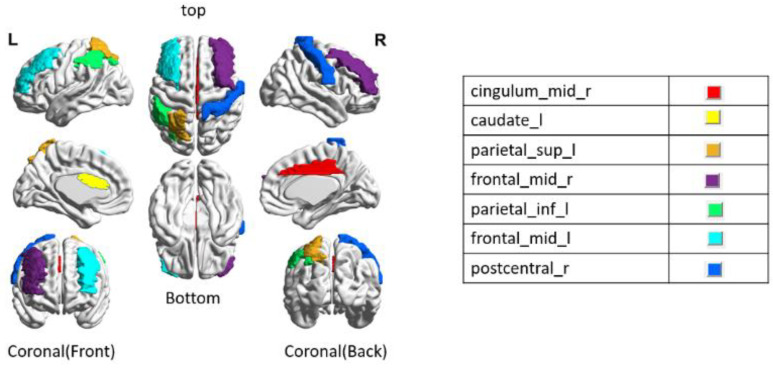
7 Overlapped discriminative ROI specifications in the AAL-90 from the three AD classifications.

**Table 1 sensors-22-08887-t001:** 10 graphic features representing the connectivity of ROIs.

Feature Name	Reference	Meaning
Degree	[24]	The number of edges incident to the vertex
Node strength	[24]	Strength of node
Diversity coefficient	[25]	Coefficient to measure the diversity of vertex
Betweenness centrality	[26]	The number of times a node acts as a bridge along the shortest path between two other nodes
K-coreness centrality	[27]	Used to identify the most important vertices within a graph use idea of K-core
Subgraph centrality	[28]	Used to identify the most important vertices within subgraph
Eigenvector centrality	[29]	A measure of the influence of a node in a network
PageRank centrality	[30]	Used to identify the most important vertices within a graph use idea of PageRank
Assortativity	[31]	Correlations between nodes of similar degree
One measure of network small-worldness	[32]	A measure of a small-world network

**Table 2 sensors-22-08887-t002:** Performance comparisons of different classifiers for three AD assessments (%).

Classifier	ZNN (Train/Test)	NEWFM (Train/Test)	SVM (Train/Test)
A-NM	98.9/97.7	88.6/87.3	86.9/83.8
N-AM	93.4/84.8	77.1/72.7	83.6/81.8
M-AN	82.6/72.7	67.6/63.8	76.1/71.7

**Table 3 sensors-22-08887-t003:** Accuracies by layers of ZNN for three AD assessments (%).

Layer	1st ZLU Layer (Train/Test)	2nd ZLU Layer (Train/Test)	Output Layer (Train/Test)
A-NM	82.9/79.7	93.7/82.8	98.9/97.7
N-AM	75.0/64.8	84.7/72.45	93.4/84.8
M-AN	64.1/59.5	73.8/64.1	82.6/72.7

**Table 4 sensors-22-08887-t004:** 16 Selected discriminative ROIs for three AD assessments.

Accuracy Rank of ROI	A-NM	N-AM	M-AN
**1**	**cingulum_mid_r**	**cingulum_mid_r**	**cingulum_mid_r**
**2**	caudate_r	caudate_r	postcentral_l
**3**	**caudate_l**	amygdala_l	**caudate_l**
**4**	**parietal_sup_l**	frontal_sup_orb_l	**parietal_sup_l**
**5**	**frontal_mid_r**	**caudate_l**	**frontal_mid_r**
**6**	**parietal_inf_l**	**parietal_sup_l**	**parietal_inf_l**
**7**	**frontal_mid_l**	lingual_l	**frontal_mid_l**
**8**	cuneus_r	parahippocampal_l	lingual_l
**9**	**postcentral_r**	**frontal_mid_r**	**postcentral_r**
**10**	cuneus_l	**parietal_inf_l**	cuneus_l
**11**	frontal_inf_oper_r	thalamus_l	frontal_inf_oper_r
**12**	frontal_inf_tri_r	frontal_inf_oper_l	frontal_inf_tri_r
**13**	temporal_inf_r	**frontal_mid_l**	temporal_inf_r
**14**	parietal_sup_r	cuneus_r	parietal_sup_r
**15**	cingulum_mid_l	insula_l	cingulum_mid_l
**16**	frontal_sup_medial_r	**postcentral_r**	thalamus_l

## Data Availability

https://adni.loni.usc.edu/data-samples/ accessed on 17 October 2022.

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
