# Peer review of "Zoom-In Neural Network Deep-Learning Model for Alzheimer’s Disease Assessments"

_sensors, 2022, doi:10.3390/s22228887_

Round 1

Reviewer 1 Report

1. Why performance is less for the classification accuracy for MCI vs AD-SCI.

2. What is the originality of this paper.

3. Can the proposed learning model be used to the assessment of other diseases? How about the applicable of this learning model?

4. Many terminologies in the article are not clearly explained and lack references, e.g. hyperbox, supervised subpattern and refined learning. It is unfriendly to readers and reduces the readability of the article. 

Author Response

  1. Why performance is less for the classification accuracy for MCI vs AD-SCI.

I mentioned in discussion section as follows :  AD and SCI are classifications in which symptoms are clearly distinguished. However, since MCI is an intermediate state between AD and SCI, the accuracy of the classifica-tion of MCI and OTHERS is lower than that of AD and SCI. [42] is another document showing that the MCI classification results are relatively low.

  1. What is the originality of this paper.

I mentioned in first paragraph of discussion section.

  1. Can the proposed learning model be used to the assessment of other diseases? How about the applicable of this learning model?

I mentioned in second paragraph of discussion section

  1. Many terminologies in the article are not clearly explained and lack references, e.g. hyperbox, supervised subpattern and refined learning. It is unfriendly to readers and reduces the readability of the article. 

The first paragraph of section 2 includes about takagi-sugeno and For others, references are intended to be substituted for explanations.

Reviewer 2 Report

1.In this paper, the authors have mentioned metacognitive learning several times, however, the introductions to metacognitive learning are lacked in section Introduction.

2. There are not enough references. No particularly recent references.

3. Justification of why this model’s accuracy is compared with SVM instead of comparing with other advanced neural network based algorithms can be given.

4. Along with benchmark performance metrics, the author should emphasize on time cost analysis.

Author Response

1.In this paper, the authors have mentioned metacognitive learning several times, however, the introductions to metacognitive learning are lacked in section Introduction.

  In introduction section, I mentioned about definition of metacognition and metacognitive information of proposed learning method

  1. There are not enough references. No particularly recent references.

  I added some recent references

  1. Justification of why this model’s accuracy is compared with SVM instead of comparing with other advanced neural network based algorithms can be given.

  I mentioned about CNN experiment of other paper in result section.  

  1. Along with benchmark performance metrics, the author should emphasize on time cost analysis.

  It is difficult to discuss the time cost because the proposed algorithm is not platformed.

Reviewer 3 Report

This manuscript is well organized and properly explains the existing literature, the proposed solution, and the assessment, which is convincing. However, I think authors should improve their manuscript.

1. The length of the paper needs to be increased. For example, the literature analysis is present but too concise, it should be enlarged. Also the qualitative/quantitative assessment against the existing works should be strengthen.

2. The future work of the proposed model should be provided in the conclusion section or discussion.

3. In addition to successful cases and numerical results, it is also a good practice to present the error cases and noise.

4. Kindly specify the limitations with real-time applicability proposed solution.

Author Response

  1. The length of the paper needs to be increased. For example, the literature analysis is present but too concise, it should be enlarged. Also the qualitative/quantitative assessment against the existing works should be strengthen.
    I enlarged paper.
  2. The future work of the proposed model should be provided in the conclusion section or discussion.

  I added future work in discussion section

  1. In addition to successful cases and numerical results, it is also a good practice to present the error cases and noise.

 I added about limitaions in discussion section

  1. Kindly specify the limitations with real-time applicability proposed solution.

I added about limitaions in discussion section